# Streptococcosis a Re-Emerging Disease in Aquaculture: Significance and Phytotherapy

**DOI:** 10.3390/ani12182443

**Published:** 2022-09-16

**Authors:** Hien Van Doan, Mehdi Soltani, Alexandra Leitão, Shafigh Shafiei, Sepideh Asadi, Alan J. Lymbery, Einar Ringø

**Affiliations:** 1Department of Animal and Aquatic Sciences, Faculty of Agriculture, Chiang Mai University, Chiang Mai 50200, Thailand; 2Science and Technology Research Institute, Chiang Mai University, 239 Huay Keaw Rd., Suthep, Muang, Chiang Mai 50200, Thailand; 3Centre for Sustainable Aquatic Ecosystems, Harry Butler Institute, Murdoch University, Perth, WA 6150, Australia; 4Department of Aquatic Animal Health, Faculty of Veterinary Medicine, University of Tehran, Tehran 1419963111, Iran; 5Environmental Sciences Center, Qatar University, Doha P.O. Box 2713, Qatar; 6Department of Food Hygiene and Quality Control, Faculty of Veterinary Medicine, Shahrekord University, Shahrekord 64165478, Iran; 7Department of Microbiology, Faculty of Veterinary Medicine, University of Tehran, Tehran 1419963111, Iran; 8Norwegian College of Fishery Science, Faculty of Bioscience, Fisheries and Economics, UiT The Arctic University of Norway, 18, 9019 Tromsø, Norway

**Keywords:** streptococcosis, phytotherapy, disease resistance, inhibitory activity, pathogenesis

## Abstract

**Simple Summary:**

Streptococcosis is an economical important bacterial disease that can seriously cause huge losses in the global aquaculture sector. In recent years studies have focused on to use extracts or essences of medicinal herbs and plants to control or treat the disease outbreaks and, in most cases the results were promising. The essential oils of the herbs or plants are more effective than the extracts and, the extracts examined have moderate efficacy in term of increasing fish survival against fish streptococcosis that could be due to the enhancement of fish immunity by the herb bio-compounds. The lack of dosage optimization, toxicity and bioavailability assays of a specific herb/plant or its bioactive compound in fish organs make it difficult to judge the validation of clinical efficacy of a particular herb/plant against fish streptococcosis, and thus, required further investigations.

**Abstract:**

Streptococcosis, particularly that caused by *S. iniae* and *S. agalactiae*, is a major re-emerging bacterial disease seriously affecting the global sustainability of aquaculture development. Despite a wide spread of the disease in aquaculture, few studies have been directed at assessing the in vitro antagonistic activity and *in vivo* efficacy of medicinal herbs and other plants against streptococcal agents. Most *in vitro* studies of plant extractives against *S. iniae* and *S. agalactiae* have found antibacterial activity, but essential oils, especially those containing eugenol, carvacrol or thymol, are more effective. Although essential oils have shown better anti-streptococcal activity in *in vitro* assays, *in vivo* bioassays require more attention. The extracts examined under in vivo conditions show moderate efficacy, increasing the survival rate of infected fish, probably through the enhancement of immunity before challenge bioassays. The available data, however, lack dosage optimization, toxicity and bioavailability assays of a specific plant or its bioactive compound in fish organs; hence, it is difficult to judge the validation of clinical efficacy for the prevention or treatment of fish streptococcosis. Despite the known bioactive compounds of many tested plants, few data are available on their mode of action towards streptococcal agents. This review addresses the efficacy of medicinal plants to fish streptococcosis and discusses the current gaps.

## 1. Introduction

Infections by *Streptococcus* species, *Streptococcus iniae*, *Streptococcus agalactiae*, *Streptococcus dysgalactiae, Streptococcus uberis, Streptococcus parauberis*, and *Streptococcus phocae*, are serious re-emerging bacterial diseases in humans and a wide range of terrestrial animals, fish, and marine mammals. Fish streptococcosis is one of the major infectious diseases in freshwater and marine aquaculture, affecting the sustainability of aquaculture development worldwide [1,2], and is also a zoonotic disease, with important food safety implications [3,4,5,6]. Even though vaccines have shown positive results, they are not adequately efficacious due to the wide heterogeneity of bacterial species/strains involved in the infections [7]. Consequently, fish streptococcal infections are often treated with various antibiotics, such as florfenicol, erythromycin, doxycycline and oxytetracycline [8,9,10,11]. However, due to re-infections by the pathogens, frequent treatments are required, causing major problems, including the accumulation of antibiotics in fish carcasses [12] and the release of drugs into aquatic ecosystems, increasing the likelihood of bacterial resistance [13,14,15,16,17].

Due to global demand for chemical-free aquaculture products [16,18], there is increasing interest in the use of dietary supplements or additives capable of improving fish health [19,20]. Medicinal herbs and other plants are potentially good alternatives to replace chemical substances in aquaculture due to numerous benefits, including improved growth performance, antioxidant activity, physiological conditions, and welfare status [21,22,23,24,25,26,27], antimicrobial and immune effects [28,29,30,31,32,33], and hepatoprotective effects [30,34]. Furthermore, medicinal plants are readily available, inexpensive, and more biodegradable compared to synthetic chemical compounds [35,36]. Consequently, numerous plants have been studied as treatment or preventative agents against fish streptococcosis (e.g., [20,37,38,39,40,41,42]). This review addresses the significance of fish streptococcosis and the potential for phytotherapy as an alternative to antibiotics, with a discussion of knowledge gaps and future research requirements. For clarity, the readers can refer to Table 1, Table 2, Table 3, Table 4, Table 5, Table 6 and Table 7 for details of the in vitro antagonistic effects and in vivo treatment efficacy of plants and plant products, and we will avoid excessive reference to these tables in the text.

## 2. The Disease

Species in the genus *Streptococcus* belong to the Order Lactobacillales (lactic acid bacteria) and are Gram-positive, spherical or ovoid, non-spore-forming, non-motile and facultative anaerobic bacteria. They have been isolated from water and from the gastrointestinal (GI) tracts of various animals, such as humans, cattle, chickens, dogs, cats, hamsters, mice, monkeys, nutria, camels, horses, sheep, goat, bottlenose dolphins, fish, frogs and seals [43,44,45,46]. Streptococcosis is the general term for a variety of diseases caused by members of the genus *Streptococcus*. In fish, streptococcosis is mostly reported for infections with *S. iniae*, *S. agalactiae* and *S. dysgalactiae* [47,48,49]. *S. iniae* is one of the leading fish pathogens in freshwater and saltwater aquaculture species, especially in warmer regions. The bacterium is β-hemolytic on 5% sheep blood agar, but it cannot be grouped by the Lancefield antigen method typically used to categorize *Streptococcus* species. *S. agalactiae* is β-hemolytic and carries the Lancefield group B antigen. *S. dysgalactiae* is mostly non-beta hemolytic, with the Lancefield group C antigen. 

Lethargy, loss of appetite, skin discoloration, exophthalmia/corneal opacity, abdominal distention and abnormal behavioral swimming are the most common clinical signs of streptococcosis in infected fish [9,47,50,51,52,53,54,55,56,57,58]. These clinical signs are not pathognomonic because they are not distinct from lactococcosis caused by *Lactococcus garvieae,* at least in some high susceptible species such as rainbow trout (*Oncorhynchus mykiss*) [59]. Other occasional macroscopical findings such as skin and fin hemorrhage, dorsal rigidity, vertebral deformity, tachypnoea and subcutaneous edema with ulceration are also reported in various degrees, but mostly in fish infected with *S. iniae*, *S. agalactiae* and *S. dysgalactiae*. The size and severity of the clinical signs are, however, varied and are dependent on a range of factors including fish species and size, bacterial virulence and health management criteria, particularly water temperature and dissolved oxygen (e.g., [9,47,60]). Internally, affected fish may show ascitic fluid in the abdominal cavity; the enlargement of liver and spleen; fibrinous pericarditis and peritonitis, hemorrhages in tissues of the brain, retrobulbar region, intestines and liver; and the congestion of the spleen and kidney in various degrees [50,56,58,61,62]. The clinical presentations are, however, known to be more severe in fish infected with capsulated strains of *S. iniae* and *S. agalactiae,* particularly in susceptible fish species such as rainbow trout and tilapia (*Oreochromis niloticus*). 

The observation of intracellular bacteria in various external and internal organs are a clear sign of generalized bacterial septicemia in infected fish, and the most common histopathological findings due to *S. iniae, S. agalactiae* and *S. dysgalactiae* have been reported in tissues of the brain, heart and eyes of affected fish with marked pericarditis, choroiditis and meningitis [47,52,55,56,62]. Affected fish can, however, develop various other pathological findings, including keratitis, hemorrhagic or granulomatous meningoencephalitis, interstitial nephritis, branchitis, splenitis, ophthalmitis, choroiditis, hepatitis, gastritis, enteritis, pancreatitis, peritonitis, skeletal muscle myositis and fasciitis, and ulcerative and hyperemic dermatitis, as well as granulomatous reactions and inflammatory responses [47,50,56,61]. In addition, other tissues including liver, kidney, spleen, heart and gill may be affected, showing necrosis and hemorrhage [9,52,55,56,62,63]. Little information is available, however, on the pathology of other streptococcal infections in fish. Macrophage infiltration in kidney, liver and muscle, focal necrosis in muscle fibers in freshwater fish infected with *S. parauberis* [57] and large numbers of vacuoles in the brain matrix of fish infected with *S. uberis* have been reported [58], but no microscopic pathology data are available for *S. phocae* infections in fish [53,64].

### 2.1. Pathogenesis

The mechanism of pathogenesis and virulence factors involved in the disease caused by streptococcal species in affected fish is not yet fully understood. After the colonization and multiplication of the external (skin, fin, gills or nares) or gastrointestinal tissues, the bacteria invade internal tissues and blood, causing a generalized bacteremia followed by a septicemic condition induced by bacterial toxins. In fish infected with *S. iniae*, infection of the central nervous system (CNS) causing meningitis can occur through the entrance of bacteria via the blood circulation system or by contaminated monocytes/phagocytes with bacterial cells, and the incidence of CNS infection was correlated with the bacterial concentration in the blood and the duration of the bacteremia [65]. As some fish infected with *S. iniae* can carry the pathogen asymptomatically with no clinical signs [66], further research required to understand the mechanism of pathogenesis in more detail.

Several virulence factors are reported in *Streptococcus* species, but these are mostly detected in strains isolated from terrestrial animals. In fish, most virulence factors have been reported for *S. iniae*, *S. agalactiae* and *S. dysgalactiae*, and scarce data are available for other streptococcal species. The capsular polysaccharides are thought to be one of the most important virulence factors, inducing resistance to phagocytosis and the host humoral immune responses [10,47,67,68,69]. The survival of pathogens at the intracellular stage can facilitate the progression from a local to a systemic infection [70,71], and virulent isolates expressing a completed polysaccharide capsule are more resistant to phagocytosis than other strains [69]. Some non-capsulated strains [67] are more virulent to fish, however, suggesting that intra-phagocytic survival may not be a primary mechanism of disease establishment in fish; thus, further investigations are required.

The enhancement of the apoptosis of infected cells may also assist disease establishment, as it can cause cell death without the release of cellular components, resulting in the suppression of the host inflammation responses [47]. Some strains of *S. iniae,* such as serotype II, have capsules with more surface antigens, which can present additional anti-phagocytic properties [67,72]. A cell surface Fc binding factor, which blocks the binding and activation of complement cascade, has also been demonstrated to be part of *S. iniae* pathogenesis [73]. Further, M-like protein for cell adhesion, phosphoglucomutase for sugar metabolism, streptolysin S (SLS) for the synthesis of the SLS structural peptide and the SLS modification protein, peptidoglycan deacetylase for peptidoglycan acetyl modification, cell envelope proteinase for the synthesis of IL-8-cleaving cysteine protease and SivR/S for the two-component transcriptional regulation system encoded by different genes (i.e., *simA*, *pgmA*, sagA–sagI, *agA, agB, PDI*, *cepI* and *SivR*/*S*) have been confirmed as virulence factors of *S. iniae* strains [67,69,74,75,76,77,78,79,80]. Furthermore, the C5a peptidase [78] and fibrinogen binding protein [81], and recently two novel virulence factors, an extracellular nuclease and a secreted nucleotidase, probably with enzymatic activities, have been identified in strains of *S. iniae* that were involved in the experimental infection of zebrafish [80].

Several virulence factors have been identified in pathogenic *S. agalactiae* strains, including pore-forming toxins [β-hemolysin/cytolysin, CAMP factor (a protein B that enlarges the area of hemolysis formed by the β-hemolysin elaborated from the bacterium)], factors for immune evasion (sialic acid capsular polysaccharide, C5a peptidase, serine protease), superoxide dismutase, D-alanylated lipoteichoic acid, adhesins, hyaluronate lyase, and methionine transport regulator [82,83,84]. Most of these factors, however, have been studied in strains recovered from infected terrestrial animals, and thus virulence factors in strains infecting fish require further research. Despite the transmission of the pathogen from mother to newborn being an important risk factor of infection and disease progression for *S. agalactiae* in humans [85], no data are available on the vertical transmission of this pathogen in susceptible oviparous or viviparous fish species. 

In *S. dysgalactiae,* the M-like protein is the most extensively studied virulence factor. This protein can opsonize both adherence to and entrance into host cells [86,87] and aids in immune evasion by inhibiting phagocytic activity and inactivating the complement cascade [87]. Adhesins encoded by different bacterial genes (*gfba*, *fnB*, *fbBA*, *fnBB*, *lmb* and *gapC*) are known to mediate binding to fibronectin [88,89,90,91], and the *gfba* gene can also assist bacterial entry into host endothelial cells and intracellular persistence [92,93]. Most adhesins, however, are recognized in strains of *S. dysgalactiae* isolated from affected human and other warm-blooded animals. In addition, protein G, a known virulent factor in *S. dysgalactiae* strains, can bind with circulating immunoglobulins and, hence, interfere with the host humoral immune response [94]. Furthermore, several toxins and secreted enzymes, including the hemolysins, streptolysin O and SLS [95,96] and superantigen *speG* [97], the streptokinase enzyme that enables the hydrolysis of fibrin and aids in bacterial spreading through tissues [87], have been identified in virulent isolates of *S. dysgalactiae*. 

Several potential virulence factors, including hyaluronic acid capsule, hyaluronidase, uberis factor, antiphagocytic factors (capsule, neutrophil toxin, M-like protein and R-like protein), plasminogen activator/streptokinase factor, surface dehydrogenase protein, CAMP factor, lactoferrin binding protein and surface adhesion molecule, have been identified in both *S. uberis* and *S. parauberis* in warm-blooded animals [98,99,100], but these factors have never been studied in strains isolated from diseased fish. Some strains of *S. parauberis* carrying capsuled polysaccharide genes have recovered from diseased fish [101], but the role of other virulent factors in the pathogenicity of these species needs further study. 

The capacity to form biofilms has been reported for *S. iniae*, *S. dysgalactiae, S. uberis* and *S. parauberis.* Biofilms can facilitate the survival and proliferation of bacteria in hostile environments, such as aquaculture recirculation bio-filtration systems [99,102,103], probably due to the bacterial extracellular production.

Virulence factors of *S. phocae* are mostly studied in marine mammals and rarely in fish species. Strains of *S. phocae* with antiphagocytic capsule ability are identified in experimental infections of Atlantic salmon (*Salmo salar*) [53,104,105]. In another study by González-Contreras et al. [106], cell-surface-related properties, including capsule detection, adhesion and hydrophobicity to fish mucus and cell lines, biofilm formation in skin mucus and serum resistance, were demonstrated in *S. phocae* isolates responsible for outbreaks in Atlantic salmon. More detailed studies of these properties are, however, required, as no mortalities or histopathological findings were seen in the fish injected with extracellular products. Other virulence factors, including fibronectin-binding proteins, the toxin SLS, genes encoding for a capsule [107] and the ability of the bacterium to invade fish and mammalian cell lines were also detected as part of *S. phocae* pathogenesis in aquatic animals, but these studies have mostly been in marine mammalians [104].

### 2.2. Disease Significance in Aquaculture

Despite the wide spread of infection in aquaculture, there are no recent estimates of annual losses by streptococcal pathogens in the industry [84]. The annual estimated losses caused by *Streptococcosis* were 150 and 250 million USD in 2000 and 2008, respectively [15,105,106]. In Iran, rainbow trout production in freshwater is remarkably high (about 180000 tons), but the annual loss through streptococcosis is estimated at around 30%, due to high water temperature in summer and poor health management [10,60,107,108]. Many fish species in freshwater, estuarine and marine environments are susceptible to *S. iniae*, and rainbow trout, yellow tail (*Seriola quinqueradiata*), Asian seabass (*Lates calcarifer*) and Nile tilapia (*Oreochromis niloticus*) are highly susceptible species (e.g., [9,47,50,105,109,110]). 

*Streptococcus agalactiae* is a globally emerging fish pathogen causing huge economic losses in many freshwater and saltwater species [44]. The bacterium is reported in rainbow trout, seabream, tilapia, yellowtail, several species of catfish and mullet, croaker (*Micropogonius undulatus*), killfish *(Menhaden* spp.) and silver pomfret (*Pampus argenteus*) [44,51,52,111,112,113,114,115].

The first outbreak by the α-hemolytic Lancefield group C *S. dysgalactiae* subsp. *dysgalactiae* was reported in vaccinated and non-vaccinated farmed amberjack/yellowtail in Japanese fish farms [116]. Later, the pathogen was detected in kingfish (*Seriola lalandi*), grey mullet (*Mugil cephalus*), basket mullet (*Liza alata*), cobia (*Rachycentron canadum*), hybrid red tilapia (*Oreochromis* sp.), pompano (*Trachinotus blochii*), white spotted snapper (*Lutjanus stellatus*), Amur sturgeon (*Acipenser schrenckii*), golden pomfret (*Trachinotus ovatus*) and Nile tilapia from Brazil, Japan, China, Malaysia, Indonesia, Taiwan and Iran [28,54,117,118,119].

The first study reporting streptococcosis infection by *S. parauberis* (formerly known as *S. uberis* type II) [120] in fish was revealed by Domeénech et al. [121]. Subsequently, the disease was reported in several species including turbot (*Scophthalmus maximus*), olive flounder, sea bass (*Sebastes ventricosus*), striped bass (*Morone saxatilis*) and ram cichlid (*Mikrogeophagus ramirezi*) [57,122,123,124,125]. More recently, *S. parauberis* classified as serotype III has been reported as the cause of streptococcosis in different turbot farms in China, and the isolates are different from those that infect flounder (*Paralichthys olivaceus*) in Japan and South Korea but similar to strains in Spain and the USA [101]. 

*S. uberis* is an important causative agent of bovine mastitis worldwide. Although [58] documented the first report of disease outbreak by *S. uberis* in mandarin fish (*Siniperca chuatsi*) in China, the first isolation and characterization of this streptococcal species from fish was reported by [119] in Iranian commercial rainbow trout farms. *S. uberis* was isolated from the gills of healthy fish [126], and thus, the report by Pourgholam et al. [119] is in doubt because they did not assess the pathogenicity of the isolated strains. 

*S. phocae* subsp. *salmonis* was first isolated from clinical specimens from harbor seal (*Phocae vitulina*) by Skaar et al. [127] before being isolated from diseased Atlantic salmon cage-farmed in Chile in the summer in 1999, with a reported mortality up to 25% [53,64]. In addition to fish, *S. phocae* has been recognized as an important pathogen of marine mammals, gray seal (*Halichoerus grypus*), ringed seal (*Phoca hispida*), Cape fur seal (*Arctocephalus pusillus pusillus*), southern sea otters (*Enhydra lutris nereis*), harbor porpoise (*Phocoena phocoena*) and other cetaceans, causing pneumonia or respiratory infection [128,129,130,131]. This pathogen has also been associated with urogenital neoplasia in Steller sea lions (*Eumatopias jubatus*) and skin abscesses in southern sea otters (*Enhydra lutris nereis*) [132,133], and Taurisano et al. [46] suggested that *S. phocae* is a serious disease in marine mammals.

Some streptococci species are serious zoonotic pathogens, with *S. iniae* causing bacteremia, cellulitis, meningitis and osteomyelitis [134], neonatal meningitis, sepsis and pneumonia caused by *S. agalactiae* [135,136] and bacteremia, lower limb cellulitis and meningitis caused by both *S. dysgalactiae* subsp. *equisimilus* and *S. dysgalactiae* subsp. *dysgalactiae* [137,138,139,140,141,142]. There are also reports of *S. uberis* in humans, although the accuracy of this identification is arguable [143]. A problem with streptococcal infections in aquaculture is that, in some outbreaks, the infected fish exhibit no clinical signs prior to death, and the mortalities are mostly due to bacterial septicemia that can involve the brain and nervous system [47,144]. In these cases, the consumption of infected fish, which appear clinically normal, can seriously affect public health. 

The immune system of aquatic animals can inevitably be suppressed by various stressors, which increases the animal’s susceptibility to pathogenic agents [47,145]. Stress and stressors, therefore play a significant role in the initiation and development of streptococcal infections. Streptococcal infections are highly stress-dependent and occur in farmed fish exposed to sub-optimal water quality parameters such as sudden fluctuations in temperature or salinity, high alkalinity (pH > 8), low dissolved oxygen concentration and increases in NH_3_ and NO_2_. Overfeeding, overstocking and overhandling can also cause outbreaks of streptococcosis with high cumulative mortality [108,146,147,148,149,150]. Mortality caused by streptococcal infections can be reduced by pathogen-free stock/larvae, separate water supplies for culture systems, reducing over-manipulation or transportation, the quarantining of newly arrived fish, reducing overcrowding, avoiding overfeeding, frequently removing dying and dead fish, and keeping excellent sanitary conditions [105]. These preventive precautions can, however, be exceedingly difficult and expensive to implement, as streptococcal agents are quite common in aquatic environments. Due to the formation of granulomatous reactions in different organs of affected fish [151], antibiotic therapy of streptococcal infections is unsuccessful [152]. Treatment by antibiotics may also increase water pollution through frequent drug administration and the release of excess chemical substances into the farm environments, causing further stress of fish. Additionally, frequent antibiotic therapy can increase the withholding period for fish carcasses, and this may interfere with the farm production scheme. 

## 3. Phytotherapy 

### 3.1. Rosemary (Rosmarinus officinalis)

Rosemary contains bioactive compounds, including camphor, a-bisabolol, 1,8-cineole, terpineol-4-ol, a-terpineol and limonene, which possess antibacterial activity [153,154]. Sixteen accessions of rosemary extracted in several solvents were inhibitory towards *S. iniae* strains, with the ethyl acetate extraction being most efficacious [155]. The ethyl acetate extract was also effective in inhibiting the growth of *S. agalactiae* strains [156]. The growth of seven strains of *S. iniae* isolated from diseased rainbow trout was inhibited by the herb's essential oil [157,158], although the efficacy of the ethanolic extract was lower than erythromycin [159], the antibiotic of choice used for the treatment of infection by *S. iniae*. At concentrations below those which affected bacterial growth (i.e., sub-minimum inhibitory concentrations (sub-MICs), the essential oil decreased the hemolytic activity of *S. iniae* supernatant and significantly suppressed the transcription of the *sag*A gene of bacterial isolates [158], suggesting that the essential oil could be useful for the control of *S. iniae* infection via the inhibition of the production of streptolysin S. 

Five days of feeding tilapia (*Oreochromis* sp.) with powdered rosemary leaves (3:17 *w*/*w*, leaf/feed) and ethyl acetate extract (1:24 *w*/*w*, extract/feed) significantly increased fish survival following *S. iniae* challenge (75% survival in treated fish compared to 49% survival in the untreated control) [155]. No significant difference was seen in the survival rate between the ethyl acetate extract and leaf powder treatments or between fish treated with rosemary and fish treated with oxytetracycline (1:199 *w*/*w*). Yilmaz et al. [160] fed tilapia (*O. mossambicus*) for 45 days with 1% rosemary leaves, obtaining a survival rate of 83.37% following challenge with *S. iniae*, whereas Gültepe et al. [161] found 73% survival with the same feeding regime; the difference in survival between these experiments can be explained by differences in the routes of challenge, bath and intraperitoneal injection. In addition, no significant difference was seen in tilapia survival (56% vs. 56%) following *S. iniae* challenge when the fish were fed herb leaves at 8% and 16%, but survival rates in both treatment dosages were comparable with oxytetracycline-treated fish (57% survival) and were significantly higher than control fish (35% survival) [156]. Fish fed with 4% leaves, however, exhibited similar survival as the control group [156]. While short feeding (8 days) of the herb leaves at 8% induced no difference in survival in the same species of tilapia challenged with *S. agalactiae* infection compared to control fish (27% vs. 24%), administration at 16% produced a higher survival rate (38%) [156]. Interestingly, when smaller fish of the same species were administered the herb leaf at either 8 or 16% in feed, treatments were ineffective against *S. agalactiae* challenge [156]. These data highlight the importance of the optimization of herb dosage and the route and duration of herb administration, as well as fish size. The effectiveness of rosemary against streptococcal infection could be in part due to an enhancement in fish innate and adaptive immune responses, as treated fish have exhibited increased lysozyme phagocytosis and leukocyte counts [161].

### 3.2. Shirazi Thyme (Zataria multiflora), Garden Thyme (Thymus vulgaris), Avishan-e-Denaii (Thymus daenensis)

Both essential oil and ethanolic extract of Shirazi thyme (*Zataria multiflora*) were inhibitory towards clinical strains of *S. iniae* obtained from diseased rainbow trout [157,162], but the essential oil exhibited more inhibitory activity than the extract, according to Soltani et al. [158]. Soltani et al. [158] showed that, when clinical isolates of *S. iniae* obtained from diseased rainbow trout were exposed to sub-MICs of Shirazi thyme essential oil, a dose-dependent reduction was exhibited in the hemolytic activity of the bacterial supernatant containing streptolysin S, and the transcription of the *sag*A gene was significantly downregulated in the treated bacterial isolates, suggesting that the essential oil may be used as an alternative to the antibiotic treatment of *S. iniae* infection. The anti-*S. iniae* activity of the ethanolic extract of Shirazi thyme was significantly stronger than other tested plants including peppermint (*Mentha piperita*), savory (*Satureja khuzistanica*) and chamomile/camomile (*Matricaria recutica*) and was comparable to erythromycin [159]. The inhibitory effect of Shirazi thyme is probably due to the diversity of the bioactive compounds in the herb [163,164,165], although few data are available to describe the exact mechanism of action.

The essential oil of thyme (*Thymus daenensis*) was more inhibitory than the ethanolic extract towards *S. iniae* activity in vitro [166]. The extract of garden thyme (*Thymus vulgaris*) was inhibitory to *S. agalactiae* [167], and its essential oil in combination with the essential oils of other plants including oregano (*Origanum vulgare*) and eucalypts (*Eucalyptus* spp.) revealed equal MIC and minimum bactericidal concentration (MBC) of 6.25 μL/mL [168], but there was no information on the effectiveness of garden thyme essential oil in isolation. 

Mozambique tilapia (*O. mossambicus*) fry fed with 1.0% garden thyme for 45 days exhibited a significantly higher survival rate than control fish after challenge with *S. iniae* infection [160]. Similarly, Gültepe et al. [161] obtained a significantly higher survival rate (78%) than control fish (39%) in tilapia fed a diet supplemented with 1.0% garden thyme powder for 45 days and then challenged with *S. iniae* infection. The inhibitory activity of garden thyme to streptococcal infection could be in part due to citraconic anhydride, 1,8-cineole and thymol. For instance, thymol and citraconic anhydride are well-known antibacterial substances [169,170], through the permeabilization and depolarization of the cytoplasmic membrane of the bacterial cell wall [171]. In addition, the plant may increase the fish immune response, as increased lysozyme and phagocytic activities plus immunocompetent cell population have been detected in treated fish [161].

### 3.3. Cumin (Cuminum cyminum, Nigella sativa)

Ethanolic extract of black cumin (*Nigella sativa*) exhibited moderate anti-*S. iniae* activity in vitro, with equal MIC and MBC values < 2 mg/mL [162]. Seventy-five days feeding Mozambique tilapia with cumin (*Cuminum cyminum*) as a feed additive at 0.5–2.0% was efficacious against *S. iniae* infection, providing 62.5–89.5% survival in treated fish vs. 40% survival in control fish. The highest survival achieved in the fish fed 1% cumin, with no adverse side effect on the fish growth [172]. In a subsequent study by Yılmaz et al. [173], 45-day feeding of the same fish species with cumin powder at 1–2% demonstrated a significantly increased resistance (61–86% survival) against *S. iniae* challenge, while there was no difference in survival between fish fed 0.5% (45%) and control fish (43%). As fish fed with 2% cumin powder had a lower survival (61%) than those fed 1% (86% survival), there appears to be an optimum for the effective control of streptococcosis. Such anti-*S. iniae* activity by cumin may be in part due to substances such as limonene, α–pinene and 1,8–cineole, as these compounds can improve the antioxidant activity and immune responses [174], although precise details of the mode of action warrant further studies.

### 3.4. Savory (Satureja bachtiarica, S. khuzistanica, S. montana)

The essential oil of Bakhtiari savory (*Satureja bachtiarica*) revealed high activity against both *S. iniae* and *S. agalactiae* strains, with the lowest MICs of 39 μg/mL and 31.2 µg/mL, respectively [166]. Aqueous extracts of savory (*Satureja montana*), obtained by the decoction method, however, exhibited moderate antagonistic activity towards *S. dysgalactiae* species. These findings were supported by the chemical analysis of the savory essential oil, which contained carvacrol (up to 62.3%), thymol (up to 40.6%) and terpinene (up to 28.3%). The ethanolic extract of savory (*Satureja khuzistanica*) has also been shown to be inhibitory to *S. iniae* [159].

### 3.5. Aloe (Aloe vera)

Essential oil of aloe (*Aloe vera)* exhibited low inhibitory activity against *S. iniae* strains originally isolated from diseased rainbow trout [175], while the anti-*S. iniae* effect of its ethanolic extract was comparable with erythromycin [176]. Aqueous and acetone extracts of aloe (*Aloe arborescens*), however, showed no anti-*S. uberis* activity [177]. Feeding tilapia (GIFT) with the herb at 0.5 to 2% improved fish immune-physiological variables, including antioxidant and hepatoprotective functions during *S. iniae* challenge, compared to untreated fish [27], but there was no information on the level of disease resistance in treated fish. One month of the feeding of rainbow trout with the ethanolic extract of aloe *(Barbados aloe)* at 1% or 1.5% produced a significant increase in disease resistance (>70% survival) with *S. iniae* challenge, compared to control fish (<50% survival) [39]. The findings were correlated with an enhancement in fish immune responses, including leukocytes, IgM, lysozyme and complement [39]. In a subsequent study by Tafi et al. [176], a ten-day oral therapeutic treatment with the ethanolic extract of aloe at 0.5, 1, and 1.5% in rainbow trout previously challenged with *S. iniae* demonstrated higher survival (about 67–73%) than control fish (about 50%). No significant differences were seen between aloe-treated fish and fish treated with erythromycin at 80 mg/kg body weight, with no toxic effects for the fish at any concentration of the herb.

### 3.6. Sage (Salvia officinalis, S. tomentosa, S. verticillate, Phlomis pungens)

The aqueous extract of balsamic sage (*Salvia tomentosa*) was moderately inhibitory to *S. agalactiae* [178], but only a weak inhibitory effect was seen with the methanolic extract of this species and the alcoholic extract of lilac sage (*S. verticillate*) [178]. In addition, the growth of *S. agalactiae* was inhibited by lilac sage (moderate with alcoholic extract), balsamic sage (weak with water and methanol extracts) and Jerusalem sage (*Phlomis pungens*) (moderate with water and weak with methanol extract) [178]. One-month feeding with garden sage (*S. officinalis*) extract at 0.5 to 1.5% in rainbow trout led to increased disease resistance against *S. iniae* challenge (about 65% survival) at 1.5% and reduced resistance at lower dosages (about 55%), compared to <50% survival in control fish, with this effect possibly be due to the enhancement of fish immune responses, including increased leukocyte population, IgM level, lysozyme and complement activity [39]. The in vitro anti-*S. iniae* activity of an ethanolic extract of garden sage (*Salvia officinalis*) was comparable with erythromycin [176], but when rainbow trout intraperitoneally infected with *S. iniae* were orally treated with 0.5 to 1.5% of the plant extract for ten days, they had only slightly better survival than control fish and significantly lower survival than fish treated with erythromycin (80 mg/kg b.w.). 

### 3.7. Myrtle (Myrtus communis, Rhodomyrtus tomentosa)

The essential oil of myrtle (*Myrtus communis*) extract revealed relatively high anti-*S. iniae* activity [166], while a leaf extract of rose myrtle (*Rhodomyrtus tomentos*) exhibited a strong inhibitory effect against both *S. iniae* and *S. agalactiae*, with the lowest MIC of 7.8 μg/mL, much higher than oxytetracycline MIC (0.1 μg/mL) [179]. The plant extract also revealed a dose-dependent bacteriostatic effect towards *S. agalactiae*. Pre-treatment of *S. agalactiae* culture with the extract at sub-MIC (0.25 MIC) did not change the susceptibility of bacterial cells to the lysozyme compared to untreated cells, but the pre-treated bacteria were more sensitive to H_2_O_2_ exposure; the population of bacteria pre-treated with the extract at 0.25 MIC and 0.125 MIC decreased from 7–8 log_10_ CFU/ ml to an undetectable level (<2 log_10_ CFU/mL) within 1–2 h, while untreated cells were able to survive under H_2_O_2_ conditions [179]. Further, pre-treatment of *S. agalactiae* with 0.25 MIC of rose myrtle leaf extract caused a significant reduction in the mortality (10%) of tilapia in an intraperitoneal challenge with the bacterium compared to 45–50% mortality in controls [179]. 

### 3.8. Clove Oil (Eucalyptus *sp*.)

While clove oil (*Syzygium aromaticum* = *Eugenia caryophyllus*) was inhibitory against *S. agalactiae* [167], its extract showed no inhibition activity [180]. Both 100% pure clove essential oil (Aromarant Co. Ltd., Rottingen, Germany) purified from the buds and commercial eugenol (>99%) (Tokyo Chemical Industry Co. Ltd., Tokyo, Japan) were inhibitory to the growth of *S. iniae* and *S. parauberis* [181] A comparison of the MBC/MIC ratios of eugenol (1:4) and clove essential oil (1:2) towards *S. iniae* isolates indicates a greater bactericidal tendency from clove essential oil, whereas eugenol had more bacteriostatic activity at lower concentrations [181]. These in vitro data clearly show that clove contains diverse bioactive substances with different abilities to inhibit the growth and multiplication of *S. iniae*.

The use of nano-emulsions of essential oils obtained from Tasmanian blue gum (*Eucalyptus globulus*) demonstrated ameliorating inhibitory activity against *S. iniae* compared to their essential oil counterparts [182], suggesting a better antibacterial function from the nano-emulsion of the herb.

### 3.9. Lavender (Lavendula angustifolia)

Lavender (Lavendular angustifolia) essential oil was shown to be a promising inhibitor against isolates of S. iniae and S. parauberis, mainly due to bacteriostatic activity [183]. One strain of *S. parauberis* was most sensitive, with MIC of 0.063% and an MIC:MBC ratio of 1:8, indicating the possibility of bacterial survival at high concentrations of lavender essential oil, although it is not able to grow at very low concentrations. The nano-emulsion oil of lavender demonstrated greater inhibitory activity than its essential oil counterpart [182], suggesting a better antibacterial function from the nano-emulsion of the plant. The mode of action of lavender oil, however, warrants further study.

### 3.10. Moshkoorak (Oliveria decumbens)

Essential oil, ethanolic extract and aromatic water of Moshkoorak (*Oliveria decumbens*) revealed inhibition against *S. iniae,* with the lowest and the highest MICs measured for the essential oil and aromatic water compounds, respectively [40]. Nile tilapias were orally treated with either the ethanolic extract or essential oil for 60 days at 0.01% exhibited similar survival (57%) following challenge with *S. iniae*, while the use of aromatic water at 0.125% or a combination of aromatic water and essential oil at 0.1% revealed slightly higher survival (64.28%). The survival rates of all treated fish were, however, higher than control fish. The antibacterial activity of this plant could be due to the high concentration (up to 52%) of bioactive substances such as carvacrol [40].

### 3.11. Garlic (Allium sativum)

The ethanolic extract of garlic *(Allium sativum*) was strongly inhibitory towards *S. iniae* and *S. agalactiae* strains [42,167], and the two-week feeding of Nile tilapia (*O. niloticus*) with garlic powder at 0.5 g and 1 g/100 g diet revealed significantly higher survival (54% at 0.5 g inclusion and 84% at 1.0 g inclusion) than control fish (10%) following a challenge infection with *S. iniae* [42]. The increased survival of garlic-treated fish that was in part due to the enhancement in fish immune responses (e.g., upregulation of IL-10 and IL-17 genes) reported by the authors. Such a positive correlation between in vitro antibacterial activity and in vivo disease resistance is promising, but the bioavailability of the garlic bioactive compounds in the fish tissues/serum merits further research to standardize the optimum dosage for the treatment of *S. iniae* infection.

### 3.12. Cinnamon (Cinnamomum *spp*.)

The ethanolic extract of Indian bay leaf (*Cinnamomum tamala*) was inhibitory to *S. iniae* strains [42], and the extract of cinnamon (*Cinnamomum verum*) showed high anti-*S. agalactiae* activity [167] that could be due to aromatic substances in the herbs. In addition, eight isolates of *S. uberis* were sensitive to both the essential oil and aqueous extract of cinnamon (*Cinnamomum cassia*), possibly due to the presence of compounds such as cinnamaldehyde, eugenol, cinnamic acid, witherhin, mucilage and diterpenes [184].

The seven-day feeding of tilapia with extract and powder of cinnamon (*C. verum*) showed 60% and 68.7% survival, respectively, in a *S. agalactiae* challenge [167]. No significant difference in survival rate was seen between treated fish and fish treated with oxytetracycline, suggesting that cinnamon may be a promising prophylactic tool against tilapia streptococcosis. In addition, the three-month feeding of tilapia with methanol extract of camphor (*Cinnamomum camphora*) revealed significantly higher survival (80%) against *S. agalactiae* infection than control fish (56.7% survival) [185].

### 3.13. Milletia Dielsiana (Spatholobus suberectus)

There are very scarce data on the efficacy of medicinal plants in combination with antibiotics against streptococcal infections in fish. A combination of extract of Milletia dielsiana (*Spatholobus suberectus*) and streptomycin sulfate displayed antibacterial synergism against *S. agalactiae* in an agar well diffusion assay [186]. Nile tilapia fed 0.52% the herb extract in combination with 0.013% streptomycin sulfate exhibited 80.5% relative survival compared to 52.6% in fish fed the extract alone, again indicating a synergistic effect of the herb and antibiotic for the treatment of *S. alaglactiae* infection in fish.

### 3.14. Allspice (Pimenta dioica)

A-50-day oral administration of various dosages of allspice (*Pimenta dioica*) to Mozambique tilapia fry revealed that the use of allspice at 10 g/kg feed not only provided the best growth performance but also the highest protection (80% survival compared to 38% in control fish) against *S. iniae* challenge [30]. Allspice is currently used in traditional folklore medicine [187], and its protective functions thought to be due to various bioactive substances including flavonoids, phenolic acids, catechins and several phenylpropanoids [188], as well as eugenol, cineol, phellandrene, caryophyllene and pimentol [189], which have both antibacterial activity and immune-enhancer properties [187].

### 3.15. Ginger (Zingiber officinale, Tetradenia riparia)

Water and acetone extract preparations of ginger bush (*Tetradenia riparia*) exhibited inhibitory effects against *S. uberis,* with a best index value of 4.22 [177]. The dietary supplementation of ginger (*Zingiber officinale*) at 1% to olive flounder for 8 weeks demonstrated a significantly higher survival rate following *S. inae* infection (66.7% survival) compared to control fish (5% survival) [41]. This improvement might be in part due to the enhancement of the immune responses of the treated fish. However, despite an immunostimulatory effect shown by Brum et al. [38], feeding essential oil of the herb at 0.5% to 1.5% to Nile tilapia for 55 days found no significant difference in survival between treated fish and control fish when challenged with *S. agalactiae* [38].

### 3.16. Oregano (Origanum vulgare)

The nano-emulsion of the essential oil of oregano (*Origanum vulgare*) demonstrated improved inhibitory activity against *S. iniae* compared to the essential oil counterpart [182], suggesting a better antibacterial function by the nano-emulsion of the herb. The MIC of the nano-emulsion was also lower than the tetracycline MIC. Such anti-*S. iniae* activity could be in part due to carvacrol, a monoterpenic phenol, one the major antibacterial compounds found in oregano.

### 3.17. Pomegranate (Punica granatum)

Using disk diffusion or micro-dilution methods, an ethanol extract of pomegranate (*Punica granatum*) revealed a relatively high anti-*S. iniae* activity, according to Ghasemi Pirbalouti et al. [166], while both aqueous and methanol extracts exhibited antagonistic activity towards *S. uberis* isolates with identical MIC and MBC values [190].

### 3.18. Isothiocyanates in Cruciferous Vegetables

All ten isothiocyanates (ITCs) present in cruciferous vegetables were inhibitory against *S. iniae* and *S. parauberis* strains isolated from olive flounder, but with different MIC and MBC values; the lowest MIC was measured for sulforaphane [191], and *S. paraurberis* was more susceptible to these ITCs than *S. iniae* [191].

### 3.19. Algae

The methanolic extract of marine brown alga (*Ecklonia cava*) and its ethyl acetate soluble fractions were better inhibitors than the *E. stolonifera* extract towards *S. iniae* and *S. parauberis* strains [192]. This could be due to phlorotannin compounds such as dieckol, which was found in the ethyl acetate soluble fraction of *E. cava* by Kim et al. [192], although Eom et al. [193] showed that the *n*-hexane soluble fraction of the extract showed the highest anti-*S. parauberis* activity, with MICs ranging from 256 to 1024 μg/mL [193]. Interestingly, the use of a combination of the hexane fraction with oxytetracycline exhibited a significant reduction up to 64-fold in the MIC value against *S. parauberis* isolates, suggesting a synergistic antibacterial activity.

### 3.20. Miscellaneous Plants

#### 3.20.1. *In Vitro* Anti-*S. iniae* and Anti-*S. parauberis* Bioassays

Ethanolic extracts of skullcaps (*Scutellaria radix*) and olive (*Olea europaea*) exhibited equal anti-*S. iniae* activity [162], and an ethanolic extract of chamomile/camomile (*Matricaria recutica*) was more active than peppermint (*Mentha piperita*) [159], probably due to differences in bioactive substances [163,164,165]. Ethanol extracts of oak (*Quercus branti*) and liquorice (*Glycyrrhiza glabra*) and essential oils of hogweed (*Heracleum lasiopetalum*)*,* tarhana herb (*Echinophora platyloba*), celeriac (*Kelussia odoratissima*) and lolopashmak (*Stachys lavandulifolia*) exhibited anti-*S. iniae* activity, although the essential oil of tarhana herb and the ethanol extract of oak showed the highest inhibitory activity [166]. The essential oil of pennyroyal (*Mentha pulegium*) was inhibitory against several clinical isolates of *S. iniae* recovered from diseased rainbow trout [175]. All aqueous, ethanolic and methanolic extracts of both neem (*Azadirachta* sp.) leaves and turmeric rhizome (*Curcuma longa*) were effective against *S. iniae* with an identical MIC of 1000 μg/mL that could be due to bioactive ingredients such as saponins, tannins, flavonoids and polysterols, which are present in both herbs [194]. Ethanolic extracts of the leaves of amla (*Phyllanthus emblica*), bohera leaves (*Terminalia bellirica*) and arjun (*Arjun coomaraswamy*) were inhibitory to *S. iniae* strains [42]. Skin mucus sampled from rainbow trout fed ethanol extract of stinging nettle (*Urtica dioica*) at 3% supplementation for 8 weeks showed enhanced antagonistic activity against *S. iniae* compared to mucus from control fish [195]. In a subsequent study, skin mucus from koi carp (*Cyprinus carpio* koi) fed ferula (*Ferula asafo etida*) powder at 20 and 25 g/kg in the diet for nine weeks exhibited anti-*S. iniae* activity [196]. In both studies, the immune responses of treated fish were enhanced, resulting in improved disease resistance to streptococcosis. Sera samples of rainbow trout intraperitoneally administered safflower (*Carthamus tinctorius*) ethanolic extract (100 mg/kg body weight) significantly increased anti-*S. iniae* activity compared to control fish [197], but no anti-*S. iniae* activity was seen in the sera of fish treated with a higher dosage (200 mg/kg body weight). Among forty natural Korean plants assessed by Kang et al. [198], only the extractions of nine inhibited the growth of *S. iniae* and *S. parauberis* strains, with the strongest inhibition with extracts of Jipsinnamul (*Agrimonia pilosa*), Dokhwal (*Aralia cordat*), N, Nadosongipul (*Phtheirospermum japonicum*), and Sumbadi (*Dystaenia takesimana*) providing the greatest anti-*S. parauberis* activity. Ethanolic extract of Korean blackberry (*Rubus coreanus*) inhibited the growth of *S. iniae* and *S. parauberis* at various concentrations (12.5–100 g/mL), with the extent of inhibition being dose-dependent [199]. The authors demonstrated that the extract was rich in phenolic compounds (48.36 ± 0.39 mg gallic acid equivalent/g), with a wide range of antibacterial activity against both Gram-positive and Gram-negative bacterial pathogens of fish. The essential oil of lemongrass (*Cymbopogon flexuosus*) exhibited strong activity against *S. iniae* and *S. parauberis* strains [200], and, based on the MBC/MIC ratio (<4), the essential oil was bactericidal for most of the tested strains. Essential oils of leaves of Manchurian fir (*Abies holophylla*), Japanese black pine (*Pinus thunbergii*), southern Japanese hemlock (*Tsuga sieboldii*), and pitch × loblolly hybrid pine (*Pinus rigitaeda*) exhibited activity against *S. parauberis*, with Japanese black pine and southern Japanese hemlock oils showing the highest and the lowest actively, respectively [201].

#### 3.20.2. *In Vivo* Anti-*S. iniae* Bioassay

Mozambique tilapia fed fenugreek (*Trigonella foenum graecum*) for 45 days exhibited 69% survival against *S. iniae* challenge, compared to 39% survival in control fish [161]. A three-month feeding trial of the same fish species with essential oil of sweet orange peel (*Citrus sinensis*) at 0.1 to 0.5% induced 46.67 to 58.33% survival compared to 13.3% in control fish, probably due to an enhancement of fish immune-physiological functions [37]. In a later study by Apraku et al. [202], Nile tilapia fed virgin coconut oil at various concentrations (0.75% to 3%) for eight weeks had significantly improved protection against *S. iniae* challenge compared to control fish. This may have been due in part to monoglycerides, lauric acid and monolaurin, which have been identified as components of coconut oil with antibacterial and immunostimulatory effects [203].

Eight-week dietary supplementation of yacon (*Smallanthus sonchifolius*) and blueberry (*Cyanococcus*) each at 1% to olive flounder resulted in significantly higher disease resistance, 66.7–76.7% survival, respectively, against *S. iniae* infection, compared to 5% survival in control fish [41], but the mode of action of this protection requires further study. The two-month feeding of rainbow trout with ethanolic extract of purple coneflower (*Echinacea purpurea*) (0.5, 1 and 1.5 g/kg diet) revealed higher survival compared to control fish only in fish fed a 1.5 g/kg diet, after challenge with *S. iniae* infection [204]. The addition of water hyacinth (*Eichhornia crassipes)* to the rainbow trout diets significantly enhanced fish survival following *S. iniae* infection, with the highest survival obtained in fish fed 1% of the plant [205]. Such protection may be in part due to diverse bioactive substances (e.g., phenolic compounds polyphenols and flavonoids) in water hyacinth that can stimulate fish immunity [205,206,207,208].

#### 3.20.3. *In Vitro* Anti-*S. agalactiae* Bioassay

Five flavonoids (morin, morin-3-O-lyxoside, morin-3-O-arabinoside, quercetin, and quercetin-3-O-arabinoside) obtained from lemon guava (*Psidium guajava*) leaves were inhibitory towards *S. agalactiae* strains, but with different degrees of inhibitory activity [209]. This suggests that leaves of lemon guava may be a candidate for treating *S. agalactiae* infection, but in vivo studies need to confirm this. The aqueous extract of green chiretta (*Andrographis paniculate*) displayed anti-*S. agalactiae* activity and feeding leaf powder or dried aqueous extract to tilapia decreased fish mortality after challenge with *S. agalactiae* [210].

Various extracts from aerial (leaf- stem), flower and root parts of six species of poppy plants (*Papaver*) including *P. macrostomum*, *P. dubium*, *P. argemone, P. bracteatum*, *P. armeniacus microstigma* and *P. chelidonium folium* exhibited anti-*S. agalactiae* effects, with ethanolic and methanolic extracts having greater efficacy than aqueous extracts [211]. Among these plants, *P. argemone* showed the highest inhibitory effect, followed by the root, aerial parts and flowers of *P. chelidonium folium*. In addition, the results revealed that the isokinolin alkaloids of the examined plants possessed an antimicrobial effect [211]. The ethanolic extract of leaves of Madeira vines (*Anredera diffusa*) was inhibitory to *S. agalactiae* strains [211], and among alcoholic and aqueous extracts from 22 species of Turkish plants, the strongest anti-*S. agalactiae* activity was obtained with an ethanolic extract of everlasting (*Helichrysum plicatum*) [178]. Furthermore, extracts of yellow waterlily (*Nuphar lutea*) with various solvents and the ethanolic extract of lydian broom (*Genista lydia*) displayed a moderate anti-*S. agalactiae* activity, but only a weak inhibitory effect was seen with extracts of European white-water lily (*Nymphaea alba*) [178].

Ethanolic extracts of orange cestrum (*Cestrum auriculatum*)*,* rhatany (*Krameria triandra*) and sauco (*Sambucus peruviana*) [212], both aqueous and alcoholic extracts of roselle (*Hibiscus sabdariffa Linn*) and alcoholic extracts of koon (*Cassia fistula* Linn) and banana (*Musca saientum Linn*) inhibited the growth of *S. agalactiae* strains [213]. Roselle was superior to the other plants in terms of toxicity and residual effects, and the alcoholic extracts of roselle were more efficacious than the aqueous extract. Aqueous and methanolic extracts of kulikhara (*Asteracantha longifolia*)*,* crowfoot grass (*Dactyloctenium indicum*) and Indian borage (*Trichodesma indicum*) were inhibitory to *S. agalactiae*, with methanolic extractions showing a better inhibition [214].

Among black pepper (*Piper nigrum*), curry leaf (*Murraya koenigii*), onion (*Allium cepa*) and Vietnamese coriander (*Persicaria odorata*), a methanolic extract of curry leaf was the most inhibitory towards *S. agalactiae* strains [180]. Aqueous extracts of drumstick tree (*Moringa oleifera*) were more inhibitory towards *S. agalactiae* biotype II than garden apple (*Aegle marmelos*) [215]. Chloroform-extracted compounds of Chinese mahogany (*Toona sinenses*) also exhibited activity against biotype II, while the ethanolic extract of the neem (margosa) tree (*Azadirachta indica*) showed moderate inhibitory activity [215], and the authors suggested drumstick tree leaves as a potential source of anti-streptococcal agents [215].

#### 3.20.4. *In Vivo* Anti-*S. agalactiae* Bioassay

The oral use of the aqueous extract of mampat (*Cratoxylum formosum)* at 0.5–1.5% in feed enhanced the innate immune responses of tilapia and significantly increased the survival (44–90%) of fish against *S. agalactiae* challenge compared to control fish (18%), with significantly higher survival at higher dosages [216]. Tilapia fed various levels (0.025–0.4%) of shrubby sophora (*Sophora flavescens*) exhibited survival rates of 47.8 to 79.9% compared to about 20% survival in control fish after challenge with *S. agalactiae* [217]. Higher survival was seen at 0.1% feed levels, due in part to enhanced immune responses including lysozyme, complement and respiratory burst measured in fish fed shrubby sophora at this level [217]. Tilapia fed papaya (*Carica papaya*) seed and asthma weed (*Euphorbia hirta*) exhibited 70% survival against *S. agalactiae* challenge [185]. The four-week feeding of Nile tilapia with the ethanolic extract of milky mangrove (*Excoecaria agallocha*) leaf at 20–50 mg/kg in the diet led to a significantly higher survival rate (40–63%) after challenge with *S. agalactiae* compared to control fish (3% survival) [6]. Greater survival was obtained at higher dosages and was correlated with enhanced fish immune responses (immunocompetent cell population, and lysozyme, respiratory burst, phagocytosis and serum bactericidal activities).

Eight weeks of feeding tilapia with an extract of Assam tea (*Camellia sinensis*) induced an enhancement in fish immune responses (lysozyme, peroxidase, alternative complement, phagocytosis and respiratory burst) and increased disease resistance against *S. agalactiae* challenge [218]. Assam tea contains bioactive compounds, including flavanonones, phenolic acids, catechins and flavonols, that can enhance animal immunity [219,220,221,222]. The four-week feeding of Nile tilapia with a mixture of traditional Chinese herbs consisting of hawthorn (*Crataegus hupehensis*)*,* dong quai (*Angelica sinensis,* and Mongolian milkvetch (*Astragalus membranaceus*) at equal ratios produced 70% survival vs. 35% survival in control fish following challenge with *S. agalactiae*, possibly due to enhancement in immune responses (lysozyme, superoxide dismutase, catalase, heat shock protein 70 and IgM) [223]. Among the aqueous extracts of ten Chinese herbal medicines fed to Nile tilapia, Dutchman’s pipe (*Aristolochia debilis*) (fruit) and Chinese ginseng (*Panax ginseng*) (leaf) exhibited the highest therapeutic efficacy (calculated based on the effective concentration in RPS of 50%) in a *S. agalactiae* challenge [224].

#### 3.20.5. *In Vitro* Anti-*S. dysgalactiae* Bioassay

Plant-derived compounds including thymol, carvacrol, eugenol and trans-cinnamaldehyde were inhibitory towards *S. dysgalactiae*, with the strongest inhibition shown by trans-cinnamaldehyde [225]. The ethanolic extracts of common purslane (*Portulaca oleracea*) and common dandelion (*Taraxacum mongolicum*) exhibited higher inhibition of *S. dysagalactiae* strains than the aqueous extracts of the same plants [226]. Methanolic extracts of ashwagandha/winter cherry (*Withania somnifera*)*,* colocynth/bitter (*Citrullus colocynthis*) and black pepper (*Piper nigrum*) at various concentrations inhibited the growth of *S. dysgalactiae* strains, with the level of activity being dose-dependent [227]. The inhibitory activity of ashwagandha and colocynth was similar and higher than that of black pepper. It has been shown that the solubility and bioactivity of plant-derived chemical compounds are largely dependent on the method of extraction, with decoction being one of the most effective methods for the extraction of bioactive compounds [228]. When *S. dysgalactiae* was challenged with billygoat weed (*Ageratum conyzoides*), betel (*Piper betle*) and turmeric (*Curcuma domestica*), all herbs inhibited growth of the bacterium even at high bacterial concentrations [229].

#### 3.20.6. *In Vitro* Anti-*S. uberis* Bioassays

There are fewer data available for phototherapeutic efficacy towards *S. uberis* than for other streptococcal species in fish. As *S. uberis* is a zoonotic pathogen, the in vitro data discussed here are for isolates recovered from terrestrial animals. Four plant-derived compounds, including trans-cinnamaldehyde (TC), eugenol, carvacrol and thymol, were inhibitory to *S. uberis* isolates, with TC exhibiting better inhibition than the others [225]. Bioactive substances obtained by supercritical fluid extraction with carbon dioxide from lichen (*Usnea barbata*) exhibited stronger anti-*S. uberis* activity than ethanolic extracts [230]. The extract with a lower usnic acid level was most inhibitory and was also more effective against *S. uberis* than ampicillin, erythromycin and penicillin [230]. Both peperina (*Minthostachys verticillate*) essential oil and limonene were antagonistic to *S. uberis* strains, with limonene having a lower MIC value [231]. The aqueous and acetone extracts of thorny acacia (*Acacia nilotica*) leaves [232] and fairy crassula (*Crassula multicava*) [177] were inhibitory to *S. uberis* strains. The inhibitory activity of thorny acacia was dose-dependent and could be in part due to the bioactive compounds, including the carbohydrates, glycosides, phytosterols, phenols, saponins and flavonoids that are major constituents of this plant [232].

**Table 1 animals-12-02443-t001:** *In vitro* inhibitory activity of medicinal/plants to *S. iniae.*

Bacterial Origin/Source	Medicinal Herb/Plant	Extraction/Essence Method	Inhibitory Method	MIC or Zone of Inhibition	MBC	Temp (°C)	Ref.
Tilapia	*Rosmarinus officinalis*	Methanolic extract	Disk diffusion	4.3–17.1 mm/mg	Unknown	25	[155]
Tilapia	*Rosmarinus officinalis*	Ethanolic extract	Disk diffusion	5.7–19.7 mm/mg	Unknown	25	[155]
Tilapia	*Rosmarinus officinalis*	Methanol/ethyl acetate (1:1)	Disk diffusion	3.1–23.8 mm/mg	Unknown	25	[155]
Tilapia	*Rosmarinus officinalis*	Ethyl acetate	Disk diffusion	9.38–37.5 m/mg	Unknown	25	[155]
Rainbow trout	*Rosmarinus officinalis*	Essential oil	Microdilution	0.12–0.25 µL/mL	0.5–1.0 µL/mL	25	[157]
Rainbow trout	*Eucalyptus camaldulensis*	Essential oil	Microdilution	160–320 µL/mL	>320 µL/mL	25	[175]
Rainbow trout	*Mentha pulegium*	Essential oil	Microdilution	40–320 µL/mL	>320 µL/mL	25	[175]
Rainbow trout	*Aloe vera*	Essential oil	Microdilution	>320 µL/mL	>640 µL/mL	25	[175]
Rainbow trout	*Zataria multiflora*	Essential oil	Microdilution	0.06 µL/mL	0.12–0.5 µL/mL	25	[157]
Rainbow trout	*Zataria multiflora*	Ethanolic extract	Microdilution	0.125 mg/mL	Unknown	Unknown	[162]
Rainbow trout	*Punica granatum*	Ethanolic extract	Microdilution	0.125 mg/mL	Unknown	Unknown	[162]
Rainbow trout	*Nigella sativa*	Ethanolic extract	Microdilution	<2 mg/mL	<2 mg/mL	Unknown	[162]
Rainbow trout	*Scutellaria radix*	Ethanolic extract	Microdilution	<2 mg/mL	<2 mg/mL	Unknown	[162]
Rainbow trout	*Olea europaea*	Ethanolic extract	Microdilution	<2 mg/mL	<2 mg/mL	Unknown	[162]
*S. iniae* KCTC 3657	*Agrimonia pilosa*	Water extract	Disk diffusion	29–34 mm (1000 ppm)	Unknown	25	[198]
*S. iniae* KCTC 3657	*Aralia cordat*	Water extract	Disk diffusion	9–14 mm (3000 ppm)	Unknown	25	[198]
*S. iniae* KCTC 3657	*Quercus mongolica*	Water extract	Disk diffusion	9–14 mm	Unknown	25	[198]
*S. iniae* KCTC 3657	*Phtheirospermum japonicum*	Water extract	Disk diffusion	9–14 mm (4000 ppm)	Unknown	25	[198]
*S. iniae* KCTC 3657	*Geranium wilfordi*	Water extract	Disk diffusion	9–14 mm	Unknown	25	[198]
*S. iniae* KCTC 3657	*Carpinus laxiflora*	Water extract	Disk diffusion	9–14 mm	Unknown	25	[198]
*S. iniae* KCTC 3657	*Sedum takesimens*	Water extract	Disk diffusion	9–14 mm	Unknown	25	[198]
*S. iniae* KCTC 3657	*Dystaenia takesimana*	Water extract	Disk diffusion	24–29 mm (3000 ppm)	Unknown	25	[198]
Marine fish	*Rubus coreanus*	Ethanolic extract	Disk diffusion	7.2 ± 0.07 mm (100 µg/mL)	Unknown	35–37	[199]
Rainbow trout	*Punica granatum* (flower)	Ethanolic extract	Microdilution	>1000 μg/mL	Unknown	37	[166]
Rainbow trout	*Quercus branti* (seed)	Ethanolic extract	Microdilution	625 μg/mL	Unknown	37	[166]
Rainbow trout	*Glycyrrhiza glabra* (root)	Ethanolic extract	Microdilution	>1000 μg/mL	Unknown	37	[166]
Rainbow trout	*Heracleum lasiopetalum* (fruit)	Essential oil	Microdilution	78 μg/mL	Unknown	37	[166]
Rainbow trout	*Satureja bachtiarica* (aerial)	Essential oil	Microdilution	39 μg/mL	Unknown	37	[166]
Rainbow trout	*Thymus daenensis* (aerial plant)	Essential oil	Microdilution	312 μg/mL	Unknown	37	[166]
Rainbow trout	Myrtus communis (leaf)	Essential oil	Microdilution	>1000 μg/mL	Unknown	37	[166]
Rainbow trout	*Echinophora platyloba* (aerial)	Essential oil	Microdilution	312 μg/mL	Unknown	37	[166]
Rainbow trout	*Kelussia odoratissima* (leaf)	Essential oil	Microdilution	>1000 μg/mL	Unknown	37	[166]
Rainbow trout	*Stachys lavandulifolia* (flower)	Ethanolic extract	Microdilution	>1000 μg/mL	Unknown	37	[166]
Rainbow trout	*Rosmarinus officinalis*	Essential oil	Microdilution	0.06 μL/mL	0.5 μL/mL	25	[158]
Olive flounder	Cruciferous vegetables	Sulforaphane	Microdilution	0.09 ± 0.03 mg/mL	0.28 ± 0.15 mg/mL	37	[191]
Olive flounder	Cruciferous vegetables	Sulforaphane	Microdilution	0.25 mg/mL	1 mg/mL	37	[191]
Olive flounder	Cruciferous vegetables	Iberin	Microdilution	0.25 mg/mL	1 mg/mL	37	[191]
Olive flounder	Cruciferous vegetables	Erucin isothiocyanates	Microdilution	0.09 ± 0.03 mg/mL	0.5 ± 0.3 mg/mL	37	[191]
Olive flounder	Cruciferous vegetables	Allyl isothiocyanates	Microdilution	>4 mg/mL	>4 mg/mL	37	[191]
Olive flounder	Cruciferous vegetables	Hexyl isothiocyanates	Microdilution	4 mg/mL	>4 mg/mL	37	[191]
Olive flounder	Cruciferous vegetables	Phenylethyl isothiocyanates	Microdilution	0.625 ± 0.25 mg/mL	1.125 ± 0.629 mg/mL	37	[191]
Olive flounder	Cruciferous vegetables	Benzyl isothiocyanates	Microdilution	0.219 ± 0.06 mg/mL	0.25 mg/mL	37	[191]
Olive flounder	Cruciferous vegetables	Phenyl isothiocyanates	Microdilution	2 mg/mL	>4 mg/mL	37	[191]
Olive flounder	Cruciferous vegetables	Indole-3-carbinol	Microdilution	0.125 mg/mL	0.25 mg/mL	37	[191]
Olive flounder	Radish root	Hydrolysate of radish root	Microdilution	0.25 mg/mL	0.25 mg/mL	37	[191]
Marine fish	*Ecklonia cava*	Methanolic extract	Disk diffusion	14.5 mm	Unknown	Unknown	[192]
Marine fish	*Ecklonia cava*	Butanol fraction	Disk diffusion	7 mm	Unknown	Unknown	[192]
Marine fish	*Ecklonia cava*	Water fraction	Disk diffusion	10 mm	Unknown	Unknown	[192]
Marine fish	*Ecklonia cava*	Ethyl acetate soluble fraction	Microdilution	256 μL/mL	Unknown	Unknown	[192]
Marine fish	*Ecklonia stolonifera*	Methanolic extract	Disk diffusion	11 mm	Unknown	Unknown	[192]
Rainbow trout	*Zataria multiflora*	Essential oil	Microdilution	0.12 μL/mL	0.25 μL/mL	25	[158]
Tilapia	*Rhodomyrtus tomentos*	Ethanolic extract	Microdilution	7.8 μg/mL	15.2–31.2 μg/mL	37	[179]
*S. iniae* ATCC2917	*Urtica dioica*	Ethanolic extract	Microdilution	200 μg/mL	Unknown	37	[195]
Unknown	*Carthamus tinctorius*	Ethanolic extract	Microdilution	Bactericidal effect	Unknown	37	[197]
African catfish	*Azadirachta indica* leaf	Aqueous extract	Well diffusion	25 mm	Unknown	37	[194]
African catfish	*Azadirachta indica* leaf	Ethanolic extract	Well diffusion	25 mm	Unknown	37	[194]
African catfish	*Azadirachta indica* leaf	Methanolic extract	Well diffusion	15 mm	Unknown	37	[194]
African catfish	Turmeric rhizome	Aqueous extract	Well diffusion	25 mm	Unknown	37	[194]
African catfish	Turmeric rhizome	Ethanolic extract	Well diffusion	15 mm	Unknown	37	[194]
African catfish	Turmeric rhizome	Methanolic extract	Well diffusion	20 mm	Unknown	37	[194]
African catfish	*Azadirachta indica* leaf	Methanolic extract	Microdilution	1000 μg/mL	Unknown	37	[194]
African catfish	Turmeric rhizome	Methanolic extract	Microdilution	1000 μg/mL	Unknown	37	[194]
Marine fish	*Origanum vulgare*	Essential oil	Microdilution	25 μg/mL	25 μg/mL	24	[182]
Marine fish	*Eucalyptus globulus*	Essential oil	Microdilution	100 μg/mL	100 μg/mL	24	[182]
Marine fish	*Melaleuca alternifolia*	Essential oil	Microdilution	100 μg/mL	100 μg/mL	24	[182]
Marine fish	*Lavendula angustifolia*	Essential oil	Microdilution	100 μg/mL	100 μg/mL	24	[182]
Marine fish	*Origanum vulgare*	Nano-emulsion of essential oil	Microdilution	12.5 μg/mL	12.5 μg/mL	24	[182]
Marine fish	*Eucalyptus globulus*	Nano-emulsion of essential oil	Microdilution	100 μg/mL	100 μg/mL	24	[182]
Marine fish	*Melaleuca alternifolia*	Nano-emulsion of essential oil	Microdilution	50 μg/mL	50 μg/mL	24	[182]
Marine fish	*Lavendular angustifolia*	Nano-emulsion of essential oil	Microdilution	100 μg/mL	100 μg/mL	24	[182]
Olive flounder	*Lavendular angustifolia*	essential oil	Microdilution	0.06–0.12% (*v*/*v*)	0.5–4.0% (*v*/*v*)	27	[183]
Olive flounder	*Syzygium aromaticum*	essential oil	Microdilution	0.25–0.5% *v*/*v*	0.25–1% *v*/*v*	27	[181]
Olive flounder	*Syzygium aromaticum*	Eugenol	Microdilution	0.125–0.5% *v*/*v*	0.5–1% *v*/*v*	27	[181]
Unknown	*Oliveria decumbens*	Ethanolic extract	Microdilution	18.75 mg/mL	75 mg/mL	25	[40]
Unknown	*Oliveria decumbens*	Essential oil	Microdilution	0.5 mg/mL	2 mg/mL	25	[40]
Unknown	*Oliveria decumbens*	Aromatic water	Microdilution	4 mg/mL	16 mg/mL	25	[40]
Tilapia	*Allium sativum*	Ethanolic extract	Disk diffusion	13 mm	Unknown	Unknown	[42]
Tilapia	*Phyllanthus emblica*	Ethanolic extract	Disk diffusion	9 mm	Unknown	Unknown	[42]
Tilapia	*Terminalia bellirica*	Ethanolic extract	Disk diffusion	7 mm	Unknown	Unknown	[42]
Tilapia	*Syzygium aromaticum*	Ethanolic extract	Disk diffusion	7 mm	Unknown	Unknown	[42]
Tilapia	*Arjun coomaraswamy*	Ethanolic extract	Disk diffusion	7 mm	Unknown	Unknown	[42]
Tilapia	*Cinnamomum tamala*	Ethanolic extract	Disk diffusion	7 mm	Unknown	Unknown	[42]
*S. iniae* ATCC29178	*Ferula asafoetida*	Powder	Disk diffusion	9 mm	Unknown	37	[196]
Olive flounder	*Cymbopogon flexuosus*	Essential oil	Microdilution	0.03–0.12% (*v*/*v*)	0.125–0.5% *v*/*v*	27	[200]
Unknown	*Mentha piperita*	Ethanolic extract	Disk diffusion	18.5 mg/mL	18.5 mg/mL	37	[159]
Unknown	*Satureja khuzistanica*	Ethanolic extract	Disk diffusion	10.8 mg/mL	10.8 mg/mL	37	[159]
Unknown	*Matricaria recutica*	Ethanolic extract	Disk diffusion	8.2 mg/mL	16.5 mg/mL	37	[159]
Unknown	*Zataria multiflora*	Ethanolic extract	Disk diffusion	4.8 mg/mL	9.7 mg/mL	37	[159]
Unknown	*Rosmarinus officinalis*	Ethanolic extract	Disk diffusion	16.8 mg/mL	33.6 mg/mL	37	[159]
*S. inaie* BCG/LMG 3740	*Aloe vera*	Essential oil	Disk diffusion	4.06 mg/mL	4.06 mg/mL	37	[159]
*S. inaie* BCG/LMG 3740	*Salvia officinalis*	Ethanolic extract	Disk diffusion	2.59 mg/mL	5.18 mg/mL	37	[139]

**Table 2 animals-12-02443-t002:** *In vivo* disease resistance of medicinal/plants to *S. iniae* infection. IP = intraperitoneal injection, IM = intramuscular injection.

Medicinal Herb/Plant	Extraction Method	Fish Species	Dosage and Duration	Water Temp. (°C)	Challenge Route	Survival Rate (%)	Ref.
Rosmarinus officinalis	Leaves	Tilapia	3:17 *w/w* (leaf/feed), 5 days	26 ± 1	IP	75	[155]
*Rosmarinus officinalis*	Ethyl acetate extract	Tilapia	1:24 *w*/*w* (extract/feed), 5 days	26 ± 1	IP	80	[155]
*Rosmarinus officinalis*	Leaves	Tilapia	4%, 5 days	26 ± 1	IP	35	[156]
*Rosmarinus officinalis*	Leaves	Tilapia	8%, 5 days	26 ± 1	IP	56	[156]
*Rosmarinus officinalis*	Leaves	Tilapia	16%, 5 days	26 ± 1	IP	50	[156]
*Cuminum cyminum*	Seed meal	Tilapia	0.5–2% feed, 75 days	28.6 ± 0.1	Bath	62.5–89.5	[21]
*Trigonella foenum graecum*	Powder	Tilapia	1% in feed, 45 days	28.4 ± 0.1	Bath	84.72	[160]
*Thymus vulgaris*	Powder	Tilapia	1% in feed, 45 days	28.4 ± 0.1	Bath	86.11	[160]
*Rosmarinus officinalis*	Powder	Tilapia	1% in feed, 45 days	28.4 ± 0.1	Bath	83.37	[160]
*Cuminum cyminum*	Seed meal	Tilapia	1–2% in feed, 45 days	28.3 ± 0.1	Bath	61–84	[173]
*Thymus vulgaris*	Powder	Tilapia	1% in feed, 45 days	28.4 ± 0.6	IP	78	[161]
*Rosmarinus officinalis*	Powder	Tilapia	1% in feed, 45 days	28.4 ± 0.6	IP	73	[161]
*Trigonella foenum graecum*	Powder	Tilapia	1% in feed, 45 days	28.4 ± 0.6	IP	69	[161]
*Pimenta dioica*	Seed meal powder	Tilapia	5, 10, 15, 20 g/kg feed, 50 days	28.4 ± 0.7	Bath	49–80%	[30]
*Citrus sinensis*	Essential oil	Tilapia	0.1%, 0.3%, 0.5%, 90 days	28	Bath	46.7–58.3	[37]
Virgin coconut oil	Coconut oil	Tilapia	3% in feed, 8 weeks	28–29	IP	about 73	[48]
Virgin coconut oil	Coconut oil	Tilapia	0.75% in feed, 8 weeks	28–29	IP	about 60	[48]
Virgin coconut oil	Coconut oil	Tilapia	1.5% in feed, 8 weeks	28–29	IP	about 67	[48]
Virgin coconut oil	Coconut oil	Tilapia	2.25% in feed, 8 weeks	28–29	IP	about 54	[48]
*Aloe vera* (Barbados aloe)	Ethanolic extract	Rainbow trout	1%, 1.5%, 30 days	14 ± 1	IP	76	[39]
*Salvia officinalis* (Sage)	Ethanolic extract	Rainbow trout	1.5%, 30 days	14 ± 1	IP	65	[39]
Yacon *(Smallanthus sonchifolius)*	Powder	Olive flounder	1%, 56 days	18 ± 3	IP	76.7	[41]
Ginger *Zingiber officinale*	Powder	Olive flounder	1%, 56 days	18 ± 3	IP	66.7	[41]
Blueberry *(Cyanococcus**)*	Powder	Olive flounder	1%, 56 days	18 ± 3	IP	85	[41]
*Oliveria decumbens*	Essential oil	Tilapia	0.01%, 60 days	Unknown	Oral	57	[40]
*Oliveria decumbens*	Ethanolic extract	Tilapia	0.01%, 60 days	Unknown	Oral	57.14	[40]
*Oliveria decumbens*	Aromatic water	Tilapia	0.125%, 60 days	Unknown	Oral	64.28	[40]
*Oliveria decumbens*	Extract + essential oil	Tilapia	0.1%, 60 days	Unknown	Oral	64.28	[40]
*Allium sativum*	Ethanolic extract	Tilapia	0.5 g/100 g feed,	Unknown	IM	~54	[42]
*Allium sativum*	Ethanolic extract	Tilapia	1 g/100 g feed,	Unknown	IM	~84	[42]
*Eichhornia crassipes*	Aqueous extract	Rainbow trout	0.25, 0.5, 1% in feed, 56 days	15 ± 1.1	IP	21.8 ± 6.5–34.7 ± 14.3	[205]
*Eichhornia crassipes*	Methanolic extract	Rainbow trout	0.25, 0.5, 1%, in feed, 56 days	15 ± 1.1	IP	24.8 ± 7.3–49.6 ± 4.7	[205]
*Salvia officinalis*	Ethanolic extract	Rainbow trout	0.5, 1, 1.5% in feed, 10 days	14 ± 1	IP	48–58	[176]

**Table 3 animals-12-02443-t003:** *In vitro* inhibitory activity of medicinal/plants towards *S. agalactiae*.

*S. agalactiae* Origin/Source	Medicinal Herb/Plant	Extraction Method	Inhibitory Method	MIC	MBC	Temp (°C)	Ref.
Strain ATCC 13813	*Cestrum auriculatum*	Ethanol extract	Agar diffusion	>0.7 cm	Unknown	37	[212]
Strain ATCC 13813	*Krameria triandra*	Ethanol extract of root/ stem	Agar diffusion	>0.7 cm	Unknown	37	[212]
Strain ATCC 13813	*Sambucus peruviana*	Ethanol extract of leaf/shoot	Agar diffusion	>0.7 cm	Unknown	37	[212]
Strain ATCC 13813	*Anredera diffusa*	Methanol	Agar diffusion	>0.7 cm	Unknown	37	[212]
Tilapia	*Cassia fistula*	Methanol	Microdilution	24.9 mg/mL	99.6 mg/mL	37	[213]
Freshwater fish	*Psidium guajava*	Methanol (morin flavonoid)	Microdilution	300 μg/mL		37	[209]
Freshwater fish	*Psidium guajava*	Methanol (morin-3-O-lyxoside flavonoid)	Microdilution	200 μg/mL	Unknown	37	[209]
Freshwater fish	*Psidium guajava*	Methanol (morin-3-O-arabinoside flavonoid)	Microdilution	150 μg/mL	Unknown	37	[209]
Freshwater fish	*Psidium guajava*	Methanol (quercetin flavonoid)	Microdilution	200 μg/mL	Unknown	37	[209]
Freshwater fish	*Psidium guajava*	Methanol (quercetin-3-O-Arabinoside flavonoid)	Microdilution	200 μg/mL	Unknown	37	[209]
Tilapia	*Hibiscus sabdariffa*	Water and methanol extract	Microdilution	4.7 mg/mL	9.4 mg/mL	37	[213]
Tilapia	*Allium sativum*	Water extract	Swab paper disc	>500 μg/mL	Unknown	25	[210]
Tilapia	*Allium sativum*	Ethanol extract	Swab paper disc	125 μg/mL	Unknown	25	[210]
Tilapia	*Allium sativum*	Methanol extract	Swab paper disc	500 μg/mL	Unknown	25	[210]
Tilapia	*Andrographis paniculata*	Water extract	Swab paper disc	31.25 μg/mL	Unknown	25	[216]
Tilapia	*Andrographis paniculata*	Ethanol extract	Swab paper disc	250 μg/mL	Unknown	25	[216]
Tilapia	*Andrographis paniculata*	Methanol extract	Swab paper disc	250 μg/mL	Unknown	25	[216]
Tilapia	*Cassia alata*	Water extract	Swab paper disc	500 μg/mL	Unknown	25	[216]
Tilapia	*Cassia alata*	Ethanol extract	Swab paper disc	250 μg/mL	Unknown	25	[216]
Tilapia	*Cassia alata*	Methanol extract	Swab paper disc	500 μg/mL	Unknown	25	[216]
Tilapia	*Garcinia mangostana*	Water extract	Swab paper disc	500 μg/mL	Unknown	25	[216]
Tilapia	*Garcinia mangostana*	Ethanol extract	Swab paper disc	250 μg/mL	Unknown	25	[216]
Tilapia	*Garcinia mangostana*	Methanol extract	Swab paper disc	500 μg/mL	Unknown	25	[216]
Tilapia	*Psidium guajava*	Water extract	Swab paper disc	500 μg/mL	Unknown	25	[216]
Tilapia	*Psidium guajava*	Ethanol extract	Swab paper disc	62.5 μg/mL	Unknown	25	[216]
Tilapia	*Psidium guajava*	Methanol extract	Swab paper disc	500 μg/mL	Unknown	25	[216]
Tilapia	*Streblus asper*	Water extract	Swab paper disc	125 μg/mL	Unknown	25	[216]
Tilapia	*Streblus asper*	Ethanol extract	Swab paper disc	250 μg/mL	Unknown	25	[216]
Tilapia	*Streblus asper*	Methanol extract	Swab paper disc	250 μg/mL	Unknown	25	[216]
Strain 55118	*Helichrysum plicatum*	Ethanolic extract	Disk diffusion	>13 mm	Unknown	37	[178]
Strain 55118	*Nuphar lutea*	Water and ethanolic extracts	Disk diffusion	Moderate	Unknown	37	[178]
Strain 55118	*Salvia tomentosa*	Water extract	Disk diffusion	Moderate	Unknown	37	[178]
Strain 55118	*Genista lydia*	Ethanolic extract	Disk diffusion	Moderate	Unknown	37	[178]
Strain 55118	*Nymphaea alba*	Water, ethanol, methanol	Disk diffusion	Weak	Unknown	37	[178]
Strain 55118	*Salvia verticillata*	Methanol and ethanol extracts	Disk diffusion	Moderate	Unknown	37	[178]
Strain 55118	*Phlomis pungens*	Water extract	Disk diffusion	Moderate	Unknown	37	[178]
Strain 55118	*Vinca minor*	Ethanolic extract	Disk diffusion	Strong	Unknown	37	[178]
Strain 55118	*Filipendula ulmaria*	Water extract	Disk diffusion	Weak	Unknown	37	[178]
Hybrid striped bass	*Rosmarinus officinalis*	Ethyl acetate	Disk diffusion	17 mm	Unknown	25	[156]
Tilapia	*Cinnamomum verum*	Water extract	Well diffusion	0.15 mg/mL	Unknown	35	[167]
Tilapia	*Allium sativum*	Water extract	Well diffusion	2.50 mg/mL	Unknown	35	[167]
Tilapia	*Eugenia caryophyllus*	Water extract	Well diffusion	0.3 mg/mL	Unknown	35	[167]
Tilapia	*Thymus vulgaris*	Water extract	Well diffusion	0.6 mg/mL	Unknown	35	[167]
Strain RITCC1913	*Papaver chelidonium folium*	Ethanol extract	Well diffusion	6.25 mg/mL	Unknown	37	[211]
Strain RITCC1913	*Papaver armeniacus microstigma*	Ethanol extract	Well diffusion	6.25 mg/mL	Unknown	37	[211]
Strain RITCC1913	*Papaver bracteatum*	Ethanol extract	Well diffusion	6.25 mg/mL	Unknown	37	[211]
Strain RITCC1913	*Papaver argemone*	Ethanol extract	Well diffusion	0.75 mg/mL	Unknown	37	[211]
Strain RITCC1913	*Papaver dubium*	Ethanol extract	Well diffusion	3.125 mg/mL	Unknown	37	[211]
Strain RITCC1913	*Papaver macrostomum*	Ethanol extract	Well diffusion	1.56 mg/mL	Unknown	37	[211]
Unknown	*Dactyloctenium indicum*	Methanol at 100 mg/mL	Disk diffusion	10 mm	Unknown	37	[214]
Unknown	*Dactyloctenium indicum*	Aqueous extract at 100 mg/mL	Disk diffusion	9.7 mm	Unknown	37	[214]
Unknown	*Trichodesma indicum*	Methanol at 100 mg/mL	Disk diffusion	15.8 mm	Unknown	37	[214]
Unknown	*Asteracantha longifolia*	Methanol at 200 mg/mL	Disk diffusion	9 mm	Unknown	37	[214]
Unknown	*Murraya koeinigii*	Methanol extract	Microdilution	0.39 mg mL	Unknown	Unknown	[180]
Biotype 2 (Unknown)	*Aegle marmelos*	Water extract	Well diffusion	5 mg/mL	Unknown	35	[215]
Biotype (Unknown)	*Emblica officinalis*	Water extract	Well diffusion	0.6 mg/mL	Unknown	35	[215]
Biotype 2 (Unknown)	*Moringa oleifera*	Water extract	Well diffusion	0.6 mg/mL	Unknown	35	[215]
Biotype 2 (Unknown)	*Azadirachta indica*	Chloroform extract	Well diffusion	10 mg/mL	Unknown	35	[215]
Biotype 2-Unknown	*Azadirachta indica*	Ethanol extract	Well diffusion	1.25 mg/mL	Unknown	35	[215]
Biotype 2 (Unknown)	*Toona sinensis*	Chloroform extract	Well diffusion	0.15 mg/mL	Unknown	35	[215]
Biotype 2 (Unknown)	*Toona sinensis*	Ethanol extract	Well diffusion	0.6 mg/mL	Unknown	35	[215]
Strain DMST 17129	*Rhodomyrtus tomentosa*	Ethanol extract	Microdilution	62.5 μg/mL	250 μg/mL	37	[179]
Tilapia	*Rhodomyrtus tomentosa*	Ethanol extract	Microdilution	31.2–62.5 μg/mL	1000 μg/mL	37	[179]

**Table 4 animals-12-02443-t004:** *In vivo* disease resistance of medicinal/plants towards *S. agalactiae* infection. IP= intraperitoneal injection; IM = intramuscular injection.

Medicinal Herb/Plant	Extraction Method	Fish Species	Dosage and Duration	Water Temp. (°C)	Route of Challenge	Survival Rate (%)	Ref.
*Andrographis paniculata*	Aqueous	Tilapia	At ratios of 4:36 and 5:35 (*w*/*w*) in feed,	25	IP	100	[210]
*Rosmarinus officinalis*	Leaves	Tilapia	8% in feed, 8 days	26 ± 1	IP	27	[156]
*Rosmarinus officinalis*	Leaves		16% in feed, 8 days	26 ± 1	IP	38	[156]
*Cinnamomum verum*	Powder	Tilapia	In ratios of 1:20, 2:18, 3:316 in feed, 7 days	Unknown	IP	68.7	[167]
*C. verum*	Aqueous	Tilapia	In ratios of 1:30, 2:28, 3:16 (*w*/*w*) in feed, 7 days	Unknown	IP	60.5	[167]
*Cratoxylum formosum*	Aqueous	Tilapia	0.5–1.5% in feed, 30 days	56 ± 2	IP	44–90	[216]
*Sophora flavescens*	Ethanol	Tilapia	0.025–0.4% in feed, 30 days	28 ± 2	IP	47.8–79.9	[217]
*Cinnamomum camphora*	Methanol	Tilapia	2 g/kg feed, 90 days	27–29	IM	80	[185]
*Carica papaya seed*	Methanol	Tilapia	2 g/kg feed, 90 days	27–29	IM	70	[185]
*Euphorbia hirta*	Methanol	Tilapia	2 g/kg feed, 90 days	27–29	IM	70	[185]
*Zingiber officinale*	Essential oil	Tilapia	0.5% in feed, 55 days	26.70 ± 1.17	Gavage	100	[38]
*Rhodomyrtus tomentosa*	Ethanol	Tilapia	Pre-treated *S. agalactiae* at 0.25 × MIC (31.2–62.5 μg mL)	30	IP	90	[179]
*R. tomentosa*	Ethanol	Tilapia	Pre-treated S. agalactiae at 0.125 × MIC (31.2–62.5 μg/mL)	30	IP	55	[179]
*R. tomentosa*	Ethanol	Tilapia	Pre-treated S. agalactiae at 0.25 × MIC	30	IP	50	[179]
*Camellia sinensis*	Ethanol	Tilapia	1 g/kg feed, 56 days	28 ± 1	IP	60	[218]
*C. sinensis*	Ethanol	Tilapia	2 g/kg feed, 56 days	28 ± 1	IP	83.33	
*C. sinensis*	Ethanol	Tilapia	4 g/kg feed, 56 days	28 ± 1	IP	76.68	[218]
*C. sinensis*	Ethanol	Tilapia	8 g/kg feed, 56 days	28 ± 1	IP	66.68	[218]

**Table 5 animals-12-02443-t005:** *In vitro* inhibitory activity of medicinal/plants to *S. dysgalactiae.*

Bacterial Origin/Source	Medicinal Herb/Plant	Extraction/Essence Method	Inhibitory Method	MIC or Zone of Inhibition	MBC	Temp (°C)	Ref.
Cow mastitis	*Portulaca oleracea*	Aqueous extract	Disk diffusion	13.8–18 mm at 0.12–0.5 g/mL	Unknown	37	[226]
Cow mastitis	*Portulaca oleracea*	Ethanolic extract	Disk diffusion	14.8–19.6 mm at 0.12–0.5 g/mL	Unknown	37	[226]
Cow mastitis	*Taraxacum mongolicum*	Aqueous extract	Disk diffusion	13.8–18 at 0.12–0.5 g/mL	Unknown	37	[226]
Cow mastitis	*Cinnamomum verum*	Trans-cinnamaldehyde	Broth dilution	0.05%	0.4%	37	[225]
Cow mastitis	*Eugenia caryophillis*	Eugenol	Broth dilution	0.4%	0.4%	37	[225]
Cow mastitis	*Origanum glandulosum*	Carvacrol	Broth dilution	0.4%	0.8%	37	[225]
Cow mastitis	*Origanum glandulosum*	Thymol	Broth dilution	0.4%	0.9%	37	[225]
Cow mastitis	*Taraxacum mongolicum*	Ethanolic extract	Disk diffusion	14.8–19.6 mm at 0.12–0.5 g/mL	Unknown	37	[226]
Bovine mastitis	*Piper betle*	Ethanol	Well diffusion	22–26 cm at 12.5–100 mg/mL	Unknown	37	[229]
Bovine mastitis	*Ageratum conyzoides*	Ethanol	Well diffusion	14–17 cm at 12.5–100 mg/l	Unknown	37	[229]
Bovine mastitis	*Curcuma domestica*	Ethanol	Well diffusion	18–21 cm at 12.5–100 mg/mL	Unknown	37	[229]
Bovine mastitis	*Withania somnifera* (root)	Methanol	Well diffusion	8.86–17.5 mm at 31.25–250 mg/mL	Unknown	37	[227]
Bovine mastitis	*Citrullus colocynthis* pulp of fruit	Methanol	Well diffusion	8.83–17.33 mm at 31.25–250 mg/mL	Unknown	37	[227]
Bovine mastitis	*Piper nigrum* (fruit)	Methanol	Well diffusion	7.7–11.4 mm at 15.6–250 mg/mL	Unknown	37	[227]

**Table 6 animals-12-02443-t006:** *In vitro* inhibitory of medicinal /plants to *S. parauberis.*

Bacterial Origin/Source	Medicinal Herb/Plant	Extraction/Essence Method	Inhibitory Method	MIC or Zone of Inhibition	MBC	Temp (°C)	Ref.
*S. parauberis* KCTC 3651	*Epilobium pyrricholophum*	Water extract	Disk diffusion	9–14 mm	Unknown	25	[198]
*S. parauberis* KCTC 3651	* Aralia * *cordat*	Water extract	Disk diffusion	9–14 mm (4000 ppm)	Unknown	25	[198]
*S. parauberis* KCTC 3651	*Quercus mongolic*	Water extract	Disk diffusion	9–14 mm	Unknown	25	[198]
*S. parauberis* KCTC 3651	*Phtheirospermum japonicum*	Water extract	Disk diffusion	29–34 mm (4000 ppm)	Unknown	25	[198]
*S. parauberis* KCTC 3651	*Geranium wilfordi*	Water extract	Disk diffusion	9–14 mm	Unknown	25	[198]
*S. parauberis* KCTC 3651	*Carpinus laxiflora*	Water extract	Disk diffusion	9–14 mm	Unknown	25	[198]
*S. parauberis* KCTC 3651	*Sedum takesimense*	Water extract	Disk diffusion	9–14 mm	Unknown	25	[198]
*S. parauberis* KCTC 3651	*Dystaenia takesimana*	Water extract	Disk diffusion	19–24 mm (>5000 ppm)	Unknown	25	[198]
Marine fish	*Rubus coreanus*	Ethanol extract	Disk diffusion	7.2 ± 0.07 mm at 100 µg/mL	Unknown	35–37	[199]
Olive flounder	*Ecklonia cava*	Methanol extract	Microdilution	1024 μg/mL (11 strains)	Unknown	25	[193]
Olive flounder	*Ecklonia cava*	n-hexane soluble (Hexane) fraction	Microdilution	256–1024 μg/mL (11 strains)	Unknown	25	[193]
Olive flounder	*Ecklonia cava*	Dichloromethane fraction	Microdilution	512- > 1024 μg/mL (11 strains)	Unknown	25	[193]
Olive flounder	*Ecklonia cava*	Ethyl acetate fraction	Microdilution	512–1024 μg/mL (11 strains)	Unknown	25	[193]
Olive flounder	Cruciferous vegetables	Sulforaphane	Microdilution	0.5 mg/mL	0.87 mg/mL	37	[191]
Olive flounder	Cruciferous vegetables	Sulforaphene	Microdilution	0.125 mg/mL	1 mg/mL	37	[191]
Olive flounder	Cruciferous vegetables	Iberin	Microdilution	0.156 mg/mL	1 mg/mL	37	[191]
Olive flounder	Cruciferous vegetables	Erucin isothiocyanates	Microdilution	0.75 mg/mL	0.75 mg/mL	37	[191]
Olive flounder	Cruciferous vegetables	Allyl isothiocyanates	Microdilution	0.75 mg/mL	4 mg/mL	37	[191]
Olive flounder	Cruciferous vegetables	Hexyl isothiocyanates	Microdilution	>4 mg/mL	>4 mg/mL	37	[191]
Olive flounder	Cruciferous vegetables	Phenylethyl isothiocyanates	Microdilution	0.188 mg/mL	0.31 ± 0.12 mg/mL	37	[191]
Olive flounder	Cruciferous vegetables	Benzyl isothiocyanates	Microdilution	0.5 mg/mL	0.62 ± 0.25 mg/mL	37	[191]
Olive flounder	Cruciferous vegetable	Phenyl isothiocyanates	Microdilution	1.5 ± 0.5 mg/mL	2 ± 1.4 mg/mL	37	[191]
Olive flounder	Cruciferous vegetables	Indole-3-carbinol	Microdilution	0.375 ± 0.14 mg/mL	0.375 ± 0.14 mg/mL	37	[191]
Olive flounder	Radish root	Hydrolysate of radish root	Microdilution	0.44 ± 0.13 mg/mL	0.5 mg/mL	37	[191]
Marine fish	*Ecklonia cava*	Methanol extract	Disk diffusion	17 mm	Unknown	Unknown	[192]
Marine fish	*Ecklonia stolonifera*	Methanol extract	Disk diffusion	11 mm	Unknown	Unknown	[192]
Marine fish	*Ecklonia cava*	Ethyl acetate fraction	Microdilution	256 μg/mL	Unknown	Unknown	[192]
Marine fish	*Ecklonia cava*	Butanol fraction	Disk diffusion	9 mm	Unknown	Unknown	[192]
Olive flounder	*Lavendular angustifolia*	Essential oil	Microdilution	0.063–0.5% (*v*/*v*)	0.5–2.0% (*v*/*v*)	27	[183]
Olive flounder	*Syzygium aromaticum*	Essential oil	Microdilution	0.25–0.5% (*v*/*v*)	0.5 (*v*/*v*)	27	[181]
Olive flounder	*Syzygium aromaticum*	Eugenol	Microdilution	0.125–1.0% (*v*/*v*)	0.5–1.0% (*v*/*v*)	27	[181]
Olive flounder	*Cymbopogon flexuosus*	Essential oil	Microdilution	0.016–0.125% (*v*/*v*)	0.03–0.5% (*v*/*v*)	27	[200]
*S. parauberis* FP3287	*Abies holophylla,*	Essential oil	Disk diffusion	11 mm	Unknown	28	[201]
*S. parauberis* FP3287	*Pinus thunbergii*	Essential oil	Disk diffusion	14 mm	Unknown	28	[201]
* S. parauberis * FP3287	*Tsuga sieboldii*	Essential oil	Disk diffusion	9.75 ± 0.35 mm	Unknown	28	[201]
* S. parauberis * FP3287	*Pinus rigitaeda*	Essential oil	Disk diffusion	10.25 ± 1.77 mm	Unknown	28	[201]

**Table 7 animals-12-02443-t007:** *In vitro* inhibitory activity of medicinal /plants to *S. uberis.*

*S. uberis* Origin/Source	Medicinal Herb/Plant	Extraction/Essence	Inhibitory Method	MIC or Zone of Inhibition	MBC	Temp (°C)	Ref.
Human mouth (9 strains)	*Cinnamomum cassia* (Cinnamon bark)	Aqueous extract	Disk diffusion	2–6 mm	Unknown	37	[225]
Human mouth (9 strains)	*Cinnamomum cassia* (Cinnamon bark)	Essential oil	Disk diffusion	9 mm	Unknown	37	[225]
Cow mastitis	*Cinnamomum verum*	Trans-cinnamaldehyde	Broth dilution	0.1% *v*/*v*	0.45% *v*/*v*	39	[225]
Cow mastitis	*Eugenia caryophillis*	Eugenol	Broth dilution	0.5% *v*/*v*	0.4% *v*/*v*	39	[225]
Cow mastitis	*Origanum glandulosum*	Carvacrol	Broth dilution	0.8% *v*/*v*	1.2% *v*/*v*	39	[225]
Cow mastitis	*Origanum glandulosum*	Thymol	Broth dilution	0.6% *v*/*v*	1.4% *v*/*v*	39	[225]
Unknown	*Acacia nilotica leaf*	Hot aqueous extract	Disk diffusion	9–22 mm at 21.25–20 mg/disk	Unknown	Unknown	[232]
Cow mastitis	*Minthostachys verticillata*	Essential oil	Microdilution	14.3–114.5 mg/mL	114.5–229 mg/mL	37	[231]
Cow mastitis	Limonene	Sigma aldrich	Microdilution	3.3–52.5 mg/mL	210 mg/mL	37	[231]
Cow mastitis	*Punica granatum*	Aqueous extract	Disk diffusion	25 mm	Unknown	37	[190]
Cow mastitis	*Punica granatum*	Methanol extract	Disk diffusion	25 mm	Unknown	37	[190]
Pig	*Usnea barbata*	Supercritical carbon dioxide extraction	Microdilution	5 µg/mL	Unknown	37	[230]
Pig	*Usnea barbata*	Usnic acid	Microdilution	10 µg/mL	Unknown	37	[230]

## 4. Conclusions and Future Studies

Streptococcosis causes massive mortality and economic loss in the global aquaculture sector. Traditionally, chemotherapeutic drugs and antibiotics have been applied for reducing the impacts of streptococcal infections on aquatic organism, but this may have serious adverse effects on aquatic ecosystems, wildlife and humans. Medicinal plants and their extracts may provide more environmentally beneficial alternatives to antibiotics for the treatment and control of streptococcal infections in aquaculture. The efficacy of a large numbers of plant species against *S. iniae* and *S. agalactiae* has been assessed in vitro, but less attention has been paid to the correlation of in vitro data with in vivo clinical efficacy. Most in vivo studies have used only two susceptible fish species, tilapia and rainbow trout, and have usually found variable increased clinical efficacy against infections. The efficacy of phytotherapy depends to a substantial extent on the form of application, such as powder, essential oils or extracts. In addition, the method of inclusion, dose, duration of feeding, level of supplementation and the size of the host and species all affect the bioavailability of plants and plant products for the host species and, thereby, immunity and resistance against infection. In the case of the dietary usage of phytotherapy, it must be considered that pelleting heat can affect the functionality and the actual dose of the active ingredients. Thus, future studies are required to guarantee the safe and full efficacy of phytotherapy when included in aquafeed. Further, the mode of action and the functionality of phytotherapeutic products in the entire body needs more clarification using modern techniques (e.g., omics). Although medicinal plants have been shown to have beneficial effects on human health and have a therapeutic effect against disease agents and on various organs of the body due to their active ingredients, their toxicity evaluation in the host is crucial before being formulated for use in aquaculture.

## Data Availability

This review paper has no data availability.

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
