# Peer review of "Streptococcosis a Re-Emerging Disease in Aquaculture: Significance and Phytotherapy"

_animals, 2022, doi:10.3390/ani12182443_

Round 1

Reviewer 1 Report

Overall the paper is a thorough investigation of antibiotic activity against warm water streptococcal species and is suitable for publication given a few alterations and revisions:

The authors do a good job reviewing warm water streptococcosis , however, there is no mention of coldwater species. Cold water species include Lactococcus piscium (Williams et al., 1990), Carnobacterium maltaromaticum (synonyms C. piscicola, Lactobacillus piscicola, Collins et al., 1987; Mora et al., 2003), and Vagococcus salmoninarum (Wallbanks et al., 1990). 

This paper cant really be described as a review when coldwater species are not mentioned. 

Also there seems to changes in font sizes throughout the paper (for example Lines 225, 239, 438...)

The paper would benefit from another grammar check, and checking that scientific names are italic.

Line 548 - please confirm a MIC of 1000 g/mL?

Line 564 - what are scientific names?

Author Response

Dear Reviewer,

We would like to thank you for your time spending to review our MS and also for your valuable comments. Your comments have been includiend in the revised version and are shown with track changes. Please also kindly see the point by point responses to your comments as the attached file.

Reviewer 2 Report

The review manuscript includes the current knowledge about a disease caused by different species of Streptococcus, with special emphasis of phytotherapy, an important alternative to used antibiotics in streptococcosis treatment.

Comments:

Line 78-79: Suggested modification: 

"These clinical signs are not pathognomonic because they are not distinct from lactococcosis caused by..."

Line 200-202:  Please add which aquaculture sector in USA was affected by the losses.

Tabeles:

- Please  complete the empty fields in column 1 of the table 1.1.

- All abbreviations for example IF, IP should be explain in tables legend.

- Both Latin and English names are used in the tables (for example rainbow trout - O. mykiss, tilapia - O. niloticus). Please select one type of nomenclature in all tabeles. All Latin names should be written in italics.

- I suggest using the terms "freshwater fish" or "marine fish" in the tables instead of "fish".

Author Response

(The authors gave the same response as above.)
